# Lignin Accumulation in Three Pumelo Cultivars in Association with Sucrose and Energy Depletion

**DOI:** 10.3390/biom9110701

**Published:** 2019-11-05

**Authors:** Juan Liu, Qinghua Huang, Peizi Kang, Lei Liang, Junjia Chen

**Affiliations:** Guangdong Key Lab of Sugarcane Improvement & Biorefinery, Guangdong Provincial Bioengineering Institute (Guangzhou Sugarcane Industry Research Institute), Guangdong Academy of Sciences, Guangzhou 510316, China; ljane0505@126.com (J.L.); qinghuahuang88@126.com (Q.H.); Pansykang@126.com (P.K.)

**Keywords:** pummelo, lignin, ATP, sucrose, neutral invertase, soluble acid invertase, cell wall-bound invertase, phenylalanine ammonia-lyase

## Abstract

Lignification, which occurs in many horticultural fruit and vegetables, brings about undesirable texture and unfavorable consumer preference. However, this problem has rarely been studied. In this work, three pumelo cultivars cvs “Hongroumiyou” (HR), “Bairoumiyou” (BR), and “Huangroumiyou” (HuR) were stored at 25 °C for 90 days, and juice sacs were sampled to explore the lignin accumulation and its relationship to sucrose and energy depletion were investigated. The results displayed that HuR contained lower sucrose content, lower ATP level, but higher lignin content compared to BR and HR during postharvest storage, indicating that the sequence according to storage resistance on the basis of lignin content is as follows: HuR < BR < HR. Furthermore, sucrose degradation attributed to enhanced activities of neutral invertase (NI), soluble acid invertase (S-AI), cell wall-bound invertase (B-AI), and energy deficit on account of declined ATP level, showed significantly negative correlation with lignin accumulation, suggesting that lignin accumulation occurrence could induce sucrose degradation and energy deficit during postharvest storage. Additionally, higher activities of phenylalanine ammonia-lyase (PAL), polyphenol oxidase (PPO), peroxidase (POD) could accelerate lignin synthesis and resulted in lignin accumulation during postharvest pumelo storage.

## 1. Introduction

Pumelo (*Citrus maxima* (Burm.) Merr.) is a delicious citrus fruit widely distributed in South of Yangtze River, China and Southeast Asia. However, harvested pumelo fruit are vulnerable to juice sac granulation featured by sacs being more and more stiffened, dried, and flavor-undesirable during storage at ambient temperature. These problems can bring about quality deterioration and economic loss. Previously, it was well-documented that lignin accumulated in postharvest pumelo fruit during juice sac granulation process [1,2,3,4]. Moreover, lignin accumulation occurs not only in harvested pumelo fruit, but also in other fruit, for instance, mangosteen, loquat, kiwifruit [5,6,7]. Not only in normal ripening process, lignin accumulation occurrence has also been observed commonly in postharvest fruit and vegetable under environmental stress [8,9]. Lignin accumulation can result in heavy quality problems and great marketability loss. Therefore, it will be important to reveal the underlying mechanism of lignin accumulation, such as the major factor affecting the lignin accumulation process.

Lignin is a phenolic heteropolymer consisting three monolignols, sinapyl alcohol, coniferyl alcohol, and *p*-coumaryl alcohol which are synthesized via phenylpropanoid pathway [10,11]. Several enzymes have been found to play a critical role in lignin synthesis. Among them, PAL is involved more particularly in the cleavage of phenylalanine to cinnamic acid, PPO is mainly responsible for catalyzing cumaric acid to caffeic acid and consequently accelerate the lignin synthesis, while POD is considered as an important enzyme in the final step of catalyzing monolignols to be polymerized to lignin [12,13]. Previous studies conducted in citrus fruit indicated that increased lignin content was accompanied by enhanced activities of PAL, PPO, and POD during postharvest storage [7,13,14].

On the one hand, it has been reported that lignin synthesis was related to carbohydrate metabolism via shikimate pathway [15]. Three invertases included NI, S-AI, B-AI are thought to be the key enzymes for catalyzing the cleavage of sucrose into fructose and glucose. Hence, enhanced activities of invertases could be a symbol of increased hexose availability [16]. Previous studies conducted on transgenic tobacco indicated that overexpression of a yeast invertase could induce high accumulation of soluble sugars [17]. On the other hand, energy deficit was found to be closely related to lignin accumulation in postharvest fruit [18,19]. Our previous study showed that reduced ATP level and energy charge of harvested banana fruit could accelerate chilling injury incidence [20]. Recently, new evidence on postharvest loquat fruit have demonstrated that energy deficit played a role in lignin accumulation [21].

The purpose of this study was to investigate the differences among three pumelo cultivars during postharvest storage, in addition, lignin accumulation and its relationship to sucrose and energy depletion were also explored. Specific analyses included soluble sugars contents, ATP level, lignin content, invertases (NI, S-AI, B-AI), and enzymes (PAL, PPO, POD) related to lignin synthesis. Probable mechanism of lignin accumulation in juice sacs of pumelo fruit was also proposed.

## 2. Materials and Methods 

### 2.1. Plant Materials and Treatments

Three cultivars of pumelo fruit, “Hongroumiyou,” “Bairoumiyou,” and “Huangroumiyou” abbreviated as HR, BR, and HuR, were harvested at commercial maturity on 1 October 2018 from Dapu county, Meizhou city, Guangdong province. Fifty fruit of each cultivar was packaged with polyethylene film (0.03 mm thick) bags, and stored at 25 °C and 70%–85% relative humidity. Six fruit from each cultivar were separately taken at 0, 15, 30, 60, and 90 d, peeled and juice sacs without segment membranes were used as plant materials for analyzing sugar, energy, and lignin.

### 2.2. Determination of Sugar 

The analysis and extraction of sugar content was modified from a method put forward by previous reports [22]. Briefly, juice sacs of three cultivars were frozen in liquid nitrogen and ground into powder. A total of 3 g of powdered sample was weighed and extracted with 3 mL deionized water by vortexing for 2 min and followed by ultrasonic extraction at 40 °C for 1 h. The extract was centrifuged at 12,000× *g* for 10 min and the supernatant was subjected to HPLC analysis. Extracts were analyzed by ALTUS A-10 (Perkin Elmer) consisting of a differential refraction detector, equipped with Durashell amino column (5 μm, 4.6 × 250 mm). The column temperature was 40 °C. Injection volume was 20 μL, 70% acetonitrile and 30% deionized water were used as mobile phases at a flow rate of 1 mL min^−1^. The sugar concentration was obtained from the sugar standard curve. Sugar contents were expressed as mg g^−1^. The contents of sugars were calculated using the following equation: Sugar contents = ((C − C_0_) × V × n)/(m × 1000) × 100(1)
where C is the sugar concentration obtained from the sugar standard curve, C_0_ is the sugar concentration of blank control, and n is the dilution ratio. V is the volume and m is the weight of samples. Total soluble sugar was determined according to previously reported methods [23].

### 2.3. Assays of Enzyme Activities

Five grams of powdered juice sacs from three cultivars of pumelo were used for measuring NI, S-AI, and B-AI activities. For measuring NI activity, five grams of powdered juice sacs were extracted with 50 mL sodium phosphate solution (pH 7.8) and centrifuged at 12,000× *g* for 10 min at 4 °C. Total of 40 μL of the supernatant was mixed with 200 μL 0.1 mol L^−1^ sucrose solution (PBS (pH7.5) for the control) and heated at 37 °C for 20 min, followed by another heating at 95 °C for 10 min. The mixture was cooled down by flowing water and added with 100 μL DNS. The reaction was performed at 95 °C for 10 min, and the absorbance was measured at 540 nm. For measuring S-AI activity, the reagents and analysis procedure were the same as the NI activity except that the control was added with 200 μL 0.1 mol L^−1^ HAC-NaAC (pH 4.8). For measuring B-AI activity, five grams of powdered juice sacs were extracted with 50 mL sodium phosphate solution (pH 7.8) and centrifuged at 12,000× *g* for 10 min at 4 °C, and the sediment was retained and added with 50 mL distilled water and centrifuged at 12,000× *g* for 10 min at 4 °C, and the sediment retained was mixed with 50 mL 0.5 mol L^−1^ NaCl solution at 4 °C overnight, and centrifuged at 12,000× *g* for 20 min at 4 °C. The supernatant was mixed with 200 μL 0.1 mol L^−1^ sucrose solution (0.1 mol L^−1^ HAC-NaAC (pH 4.8) for the control) and heated at 37 °C for 20 min, followed by another heating at 95 °C for 10 min. The mixture was cooled down by flowing water and added with 100 μL DNS. The reaction was performed at 95 °C for 10 min, and the absorbance was measured at 540 nm. U g^−1^ protein was used to express the enzyme activity.

### 2.4. Determination of ATP, ADP, and AMP

Two grams of juice sacs from three cultivars of pumelo were used for measuring ATP, ADP, and AMP according to our previously reported methods [20]. µg g^−1^ fresh weight was used to express the level of ATP, ADP, and AMP.

### 2.5. Assays of PAL, POD, and PPO Activities

PAL activity was measured by a method described by Jiang and Joyce [24]. For determining POD activity, four grams of fruit juice sacs was homogenized under liquid nitrogen and 20 mL 0.2 mol L^−1^ phosphate buffer (pH 6.8) was added to it which contained 100 μmol L^−1^ EDTA, then 0.4 g PVP was added and grinded in an ice bath. After centrifugation at 15,000× *g* for 15 min at 4 °C and the supernatant was taken to measure the enzyme activity. A total of 0.05 mL of the supernatant was taken and added to a reaction solution containing 0.1 mL 4.0% guaiacol, 0.1 mL 0.46% hydrogen peroxide, and 2.75 mL phosphate buffer, the absorbance was measured at 470 nm and one unit (U) of POD was defined as 0.01 of absorbance fluctuation per minute. PPO activity was measured according to previous study [25].

### 2.6. Measurement of Lignin

A total of 1 g of powdered juice sacs from three cultivars of pumelo fruit were extracted with 20 mL 80% ethanol by vortexing for 2 min, followed by treatment in water bath at 95 °C for 10 min. The extract was cooled down by flowing water and centrifuged at 8000 rpm for 10 min. The sediment was kept and added with 15 mL 80% ethanol by vortexing for 2 min, followed by treatment in water bath at 95 °C for 10 min. After 10 min centrifugation at 8000 rpm, the sediment was obtained and extracted with 10 mL DMSO for 15 h. The extract was centrifuged at 8000 rpm for 10 min and the sediment was extracted with 10 mL acetone, followed by centrifugation at 8000 rpm for 10 min. The procedure of extraction with acetone and centrifugation was repeated three times. The sediment was lyophilized. Fifteen milligram sediment sample was weighed and mixed with 1 mL acetyl bromide, followed by treatment in water bath at 50 °C for 2 h with shaking every 15 min. The mixture was homogenized with 4 mL NaOH and 700 μL hydroxylamine hydrochloride, then diluted with 80% acetic acid to 10 mL. Absorbance was measured using a UV-Vis spectrophotometer at 280 nm. mg g^−1^ was used to express the lignin content.

### 2.7. Statistical Analysis

All parameters were analyzed three times. The values in figures and table were represented in form of the mean ± SE (standard error) (n = 3). Data was processed using the software SPSS statistics 22.0, and the results of cultivar BR were used for the correlation analysis in this study. Significant difference were examined by one-way analysis of variance. The value of *p* < 0.01 or *p* < 0.05 represented very significant difference or significant difference, respectively.

## 3. Results

### 3.1. Changes in Sugar Content in Juice Sacs of Three Pummelo Cultivars

Figure 1 showed that at harvest, BR and HuR contained high content of total soluble sugar, which reduced after 30 d of storage, increased after 60 d of storage and decreased after 90 d of storage. Total soluble sugar content of HR showed tendency of continuous enhancement until 60 d of storage, thereafter, decreased slightly (Figure 1A). Accordingly, fructose and glucose contents increased to maximum at 30 d of storage (exception, glucose of HR at 60 d), subsequently declined with the extension of storage (Figure 1B,D). Similarly, a rise from 0 d to 30 d followed by a decline from 30 d to 90 d were observed in sucrose contents of HR (Figure 1C). Unlike them, sucrose contents of BR and HuR decreased slightly during postharvest storage (Figure 1C). Compared to BR and HR, HuR contained significantly lower sucrose content during the whole storage time. The sucrose content continuously declined in juice sacs of HuR and BR during the whole storage time, while that of HR increased to a peak at 30 d and then decreased as the storage time extended. From 30 d to 60 d of storage, the sucrose content in HR was the highest, followed by BR and HuR was the lowest.

These data highlighted that faster sucrose depletion, which resulted in lower sucrose content, could be observed in juice sacs of HuR in contrast to the other two cultivars during postharvest storage.

### 3.2. Changes in NI, S-AI, B-AI Activities in Juice Sacs of Three Pummelo Cultivars

As shown in Figure 2, NI (Figure 2A), S-AI (Figure 2B), and B-AI (Figure 2C) activities showed a rising trend in pumelo juice sacs of three cultivars during postharvest storage. Compared to HuR ang BR, the activities of NI, S-AI and B-AI in juice sacs of HR were lower from 0 d to 30 d of storage. In addition, HuR showed higher B-AI activity than BR and HR from 0 d to 30 d of storage. At 30 d of storage, HuR displayed the highest NI, S-AI, and B-AI activities, followed by BR and then HR. As storage time proceeded, the activities of NI, S-AI, and B-AI increased in juice sacs of three pumelo cultivars, and BR was the highest, followed by HR and then HuR from 60 d to 90 d of storage.

These data revealed that the NI, S-AI, and B-AI activities of juice sacs of three pumelo cultivars were observed to increase as the storage time proceeded, which was suggested to contribute to sucrose degradation. Higher activities of NI, S-AI, and B-AI in juice sacs of HuR at 30 d of storage could result in lower sucrose content during the postharvest storage.

### 3.3. Changes in ATP, ADP, and AMP Levels in Juice Sacs of Three PUMMELO Cultivars

Figure 3 showed that at harvest pumelo fruit reserved high levels of ATP (Figure 3A) and ADP (Figure 3B), but relatively low AMP content (Figure 3C). As the storage time extended, ATP and ADP levels decreased, but AMP level increased. Furthermore, ADP and AMP levels decreased from 15 d to 60 d, thereafter, ADP level exhibited a sharp increase from 60 d to 90 d, while AMP level declined slowly. In particular, ATP level decreased during the whole storage. Compared to the other two cultivars, HuR contained lower level of ATP from 15 d to the rest of storage time, suggesting that cellular energy deficit in juice sacs of HuR was the most obvious in contrast to BR and HR.

These data implied that energy deficit, which was featured by declined ATP level, was one typical physiological change in juice sacs of three pumelo cultivars during postharvest storage. Cultivar of HuR maintained the lowest ATP level from 15 d to 90 d, followed by BR and HR during postharvest storage time.

### 3.4. Changes in Lignin Content in Juice Sacs of Three Pummelo Cultivars

Figure 4 illustrated that juice sacs lignin content elevated gradually before the 15 d of storage, thereafter, a sharp increase was displayed in BR and HuR from 15 d to 60 d of storage, then followed by gradual enhancement during the remaining storage. Meanwhile, juice sacs lignin content of HR exhibited relatively slower increase before 60 d of storage, but sharp increase from 60 d to 90 d of storage. Compared to HR, the lignin accumulation was higher in HuR and BR, indicating that lignin accumulation occurrence of pumelo fruit from easier to harder was HuR, BR, and HR.

These data indicated that lignin content increased as the storage time extended in juice sacs of three pumelo cultivars, and this would result in juice sacs granulation of pumelo fruit and consequently lead to loss of nutrition and commodity value. Cultivar of HuR is more vulnerable to lignin accumulation, followed by BR and then HR during postharvest storage, indicating that the storage resistance of HuR was less than BR and HR.

### 3.5. Changes in PAL, PPO, and POD Activities in Juice Sacs of Three Pummelo Cultivars

Figure 5 also displayed that PAL activity (Figure 5A) in pumelo fruit increased from 0 d to 60 d of storage, thereafter, decreased during the following storage. Meanwhile, PAL activity in HR was much higher than in HuR and BR during the whole storage. As to PPO activity (Figure 5B), which increased more drastically in HuR than HR and BR between 15 d and 60 d of storage, showed sharp decline after 60 d of storage. Moreover, juice sacs POD activity (Figure 5C) exhibited a gradual rise until 30 d, followed by a sharp increase from 30 d to 60 d in BR and HR and a slight rise in HuR, and then a sharp decline was displayed from 60 d to 90 d in three cultivars. It was worthy to be noticed that the PPO activity from 15 d to 60 d of storage was higher in the order of HuR, BR, and HR, and the same order was observed in POD activity from 0 d to 30 d.

These data indicated that increased PAL, PPO, and POD activities could contribute to lignin accumulation in juice sacs of three pumelo cultivars during postharvest storage. In addition, compared to BR and HR, higher PPO activity from 15 d to 60 d and higher POD activity from 0 d to 30 d might result in higher lignin content in juice sacs of HuR.

## 4. Discussion

### 4.1. Lignin Accumulation Was Correlated with Sugar Metabolism

Based on previous documents, sucrose acted as the core of respiration metabolism and provided precursor for various substances synthesis; therefore, lignin synthesis was closely related to sucrose content in plants. Specifically, phenylpropanoid pathway, which produces monolignols of lignin, is connected with sugar metabolism through shikimate pathway in which phenylalanine is produced [15]. It was shown that, lignin accumulated when sugar was in rapid decrease in the peach endocarp, indicating that lignin accumulation was in association with the declined sugar content, which accelerated the pit hardening process [26]. Other studies showed that total soluble sugar content increased significantly as well as lignin, and genes related to glucose and lignin synthesis were strongly induced by boron deficiency stress in leaves of citrus [27]. The possible link between lignin accumulation with sugar metabolism was reinforced by the observation that sugars availability positively affected the cell wall biosynthesis, which included lignin synthesis in *Arabidopsis thaliana* [28].

In the present study, the linearity regression analysis indicated that the increased lignin content in harvested pumelo fruit showed a significantly positive relation (r = 0.93, *p* < 0.05) (Appendix A) with total soluble sugar from 0 d to 30 d. Furthermore, the decrease in sucrose content showed a significantly negative relation with the increased lignin content (r = −0.976, *p* < 0.05) (Appendix A). Moreover, the lowest sucrose content in juice sacs of HuR and the highest lignin content from 30 d together with the higher sucrose content in juice sacs of HR from 30 d to 60 d and the lowest lignin content from 15 d verified that there was some close relationship between sucrose content and lignin accumulation, directly or indirectly. In addition, the lignin content exhibited a positive relation with glucose (r = 0.878, *p* < 0.05) (Appendix A) and fructose (r = 0.822, *p* < 0.05) (Appendix A). All these data highlighted that low sucrose content and high fructose and glucose contents were beneficial to lignin accumulation in postharvest pumelo fruit. It was worthy to be noticed that the sucrose of HR increased from 0 d to 30 d first, and then declined subsequently; however, the fructose and glucose contents increased during the whole storage (Figure 1C), therefore, the lignin accumulation was not influenced for the sake of enough carbon supply.

In our previous study, we found that higher soluble sugar content was beneficial in alleviating chilling injury of postharvest banana fruit during cold storage [20]. This suggests that soluble sugars played an unexpected role in not only normal ripening development but also in cold storage of postharvest fruit. In this work, during the normal ripening development of postharvest pumelo fruit, except for sucrose contents of HR, sucrose content decreased while glucose and fructose contents increased first and then declined slightly in juice sacs of pumelo fruit (Figure 1B–D). In addition, the activities of NI, S-AI, and B-AI kept increasing in three cultivars during the whole storage time. Moreover, the linearity regression analysis revealed that the decreased sucrose content displayed a significantly positive relation with enhanced activities of NI (r = −0.93, *p* < 0.05) (Appendix A), S-AI (r = −0.925, *p* < 0.05) (Appendix A), and B-AI (r = −0.984, *p* < 0.01) (Appendix A). Invertase plays a direct role in catalyzing irreversible degradation of sucrose to fructose and glucose. It was further reported that the invertase pathway to degrade sucrose was the main route to offer carbon supply to cell wall formation in *Arabidopsis* cell cultures [29]. Data above highlighted that the decrease in sucrose content could be attributed to the increased NI, S-AI, and B-AI activities, which might accelerate the sucrose degradation and further promote an increase in glucose and fructose contents.

Many studies proposed that lignin accumulation occurs in some other fruits, such as kiwifruit [7], pear [30], asparagus [31], mangosteen [6], loquat [8], wax apple [14], strawberry [32]. Lignin accumulation could be greatly influenced by phenolics biosynthetic pathway and cell wall metabolism during ripening stage of postharvest fruit, indicating that increased lignin content correlated with anthocyanins, flavonoids, and phenylpropanoids as well as polysaccharide and cellulose [31,33]. It has been reported that treatment of nitric oxide, methyl jasmonate, and methyl salicylate significantly inhibited the lignin deposition by enhancing PAL and POD activities in postharvest kiwifruit and wax apple fruit [7,14]. Meanwhile conversely, 1-methylcyclopropene aggravated lignin accumulation via inducing a decline in POD activity and other enzymes related to lignin synthesis [7]. PAL is a key rate-limiting enzyme responsible for catalyzation of phenylalanine to cinnamic acid, and POD mainly participates in polymerization of lignin monomer [12]. PPO could provide precursor of lignin synthesis via participating in phenols oxidation [13]. Therefore, to investigate whether the role of PAL, POD, and PPO activities are related to lignin contents, we measured the activities of PAL, POD, and PPO in juice sacs of postharvest pumelo of three cultivars (Figure 5). In addition, the linearity regression analysis of lignin content and PAL, POD, and PPO activities indicated that, the increased lignin content of harvested pumelo fruit showed a significant positive relation with PAL activity (r = 0.911, *p* < 0.05) (Appendix A), and a very significantly positive correlation with PPO (r = 0.977, *p* < 0.01) (Appendix A) and POD (r = 0.942, *p* < 0.01) (Appendix A). These results demonstrated the aggravated incidence of lignin accumulation was associated with the enhanced activities of PAL, POD, and PPO in postharvest pumelo fruit.

Taken collectively, lignin accumulation occurrence in juice sacs of BR was positively correlated with sucrose degradation, which could be attributed to increased activities of NI, S-AI, and B-AI activities. Furthermore, enhanced activities of PAL, POD, and PPO contributed to lignin synthesis and gave rise to the occurrence of lignin accumulation in juice sacs of BR. Whether the differences of the above parameters among three pumelo cultivars were statistically significant or not made no difference in the same conclusion in the other two cultivars.

### 4.2. Lignin Accumulation Was Correlated with Energy Metabolism

Sugar metabolism was not the only factor that exerted an important influence on lignin accumulation in postharvest pumelo fruit. There is also considerable evidence of a role for energy metabolism in regulating lignin accumulation. Energy plays a vital role in maintaining normal physiological activities of all living organisms. Insufficient cellular energy supply may result in metabolic disorder of harvested horticultural fruit, such as peel browning in lichi and longan [34,35], pits on blue berry [36], chilling injury of papaya and banana [20,37]. Early studies in loquat fruit indicated that energy deficit was connected closely with lignification at low temperature storage [18]. It was also previously shown that reduced lignification was involved more particularly in energy status maintenance and enhanced antioxidant enzymes activities of water bamboo shoot [32]. Moreover, It has also been reported that ATP level displayed a reverse correlation with lignification, which could be inhibited by low activities of PAL and POD, whereas, application of exogenous ATP was beneficial to protect the postharvest white mushroom against the development of lignification [19]. It was also recently shown that the relative transcriptions of genes related to lignin synthesis were effectively down-regulated, in the meantime, high energy status was maintained by an exogenous peptide treatment on postharvest loquat fruit at 0 °C [21]. In our previous study, we found that a delay in ATP decline and maintenance of high energy status was beneficial in alleviating chilling injury symptoms in postharvest banana fruit [20]. Furthermore, many studies investigated that high ATP level could protect postharvest fruit against pericarp darkening or browning incidence [21,38].

In the present work, energy deficit featured by declined ATP level became more and more obvious in postharvest pumelo fruit during postharvest storage (Figure 3A). Accordingly, the linearity regression analysis implied that lignin content displayed a very significant reverse relation with ATP (r = −0.987, *p*< 0.01) (Appendix A). This gave strong credence for a participation of ATP in lignin synthesis process probably as indicator of cell energy homeostasis in relation to metabolic activity. Moreover, results above might attributed to energy required to carry on various metabolic pathway such as the lignin accumulation process, therefore, the accumulated lignin amounts were observed at the price of continuous ATP depletion in postharvest pumelo fruit. In this meaning, it would be further speculated that exogenous treatments which are beneficial to maintain energy status might delay lignin accumulation in postharvest pumelo fruit, which is in agreement with reduced lignin accumulation under high O_2_/CO_2_ controlled atmosphere or low temperature storage as reported by previous studies [19,38]. Meanwhile conversely, those exogenous applications which accelerate energy depletion might contribute to lignin accumulation in postharvest pumelo fruit, which was in accordance with the results of our present work, because the pumelo fruit was stored at room temperature at which fruit respiration rate was high, therefore fast energy consumption could accelerate lignin accumulation.

The above results indicated that the continuously declining ATP levels can result in cellular energy deficit in juice sacs of pumelo fruit, and consequently accelerate the lignin accumulation and senescence directly or indirectly of harvested pumelo fruit.

In summary, the sucrose degradation together with ATP depletion contribute to the lignin accumulation during postharvest pumelo storage. The possible mechanism of lignin accumulation of postharvest pumelo fruit by relying on sugar and energy depletion is shown in Figure 6.

## 5. Conclusions

In this study, we compared soluble sugars contents, ATP level, lignin content, invertases (NI, S-AI, B-AI) related to sucrose depletion and enzymes (PAL, PPO, POD) related to lignin synthesis in juice sacs of HuR, BR, and HR. Compared to BR and HR, HuR contained lower sucrose content, lower ATP level, and higher lignin content. The storage resistance of HR was better than BR and HuR on account of lignin content. In addition, we revealed the aggravated lignin accumulation in juice sacs of BR was negatively correlated with the sucrose degradation which was attributed to enhanced activities of NI, S-AI and B-AI, and the energy deficit which was due to declined ATP level. Moreover, enhanced activities of PAL, PPO, and POD can accelerate the lignin accumulation in postharvest pumelo fruit.

## Figures and Tables

**Figure 1 biomolecules-09-00701-f001:**
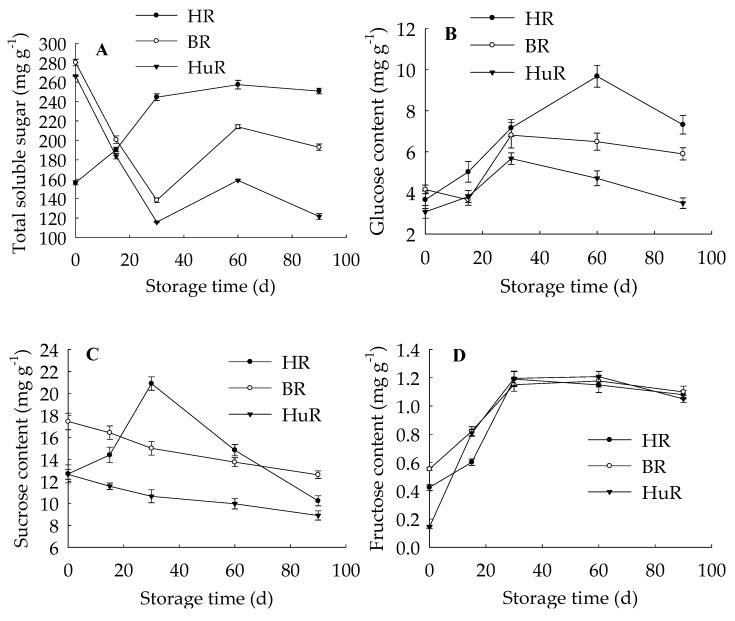
Changes in contents of total soluble sugar (**A**), glucose (**B**), sucrose (**C**), and fructose (**D**) in juice sacs of three pumelo cultivars. Vertical bars represent SE of the means of three replicate assays.

**Figure 2 biomolecules-09-00701-f002:**
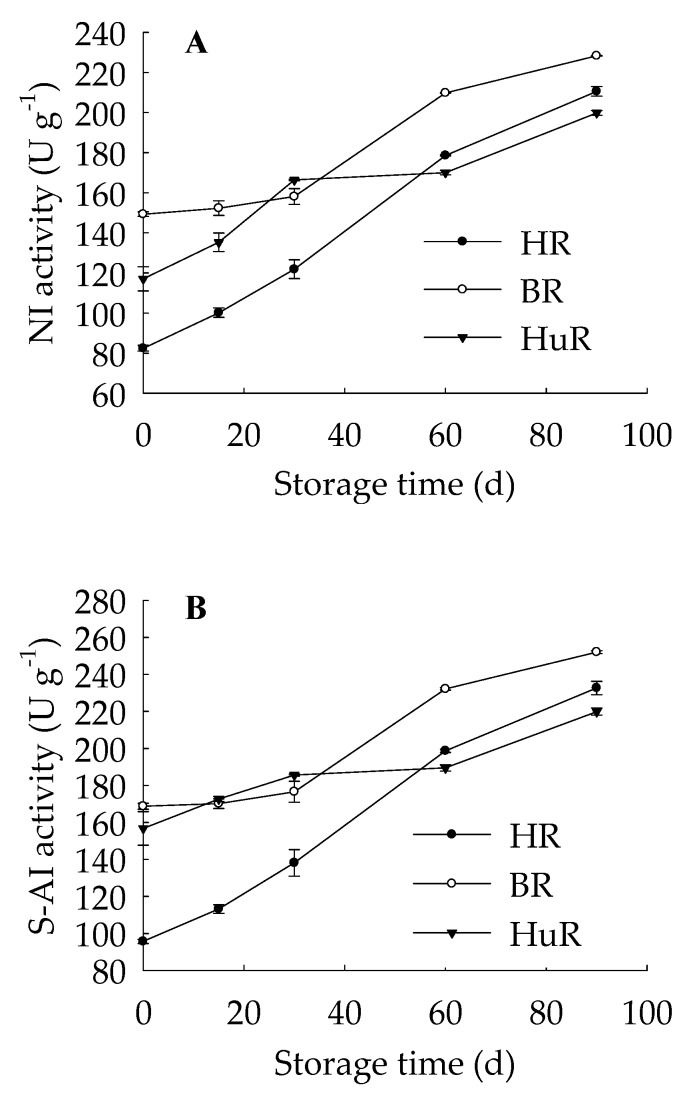
Changes in NI (**A**), S-AI (**B**), B-AI(**C**) activities in juice sacs of three pumelo cultivars. Vertical bars represent SE of the means of three replicate assays. NI, neutral invertase; S-AI, soluble acid invertase; B-AI, cell wall insoluble acid invertase.

**Figure 3 biomolecules-09-00701-f003:**
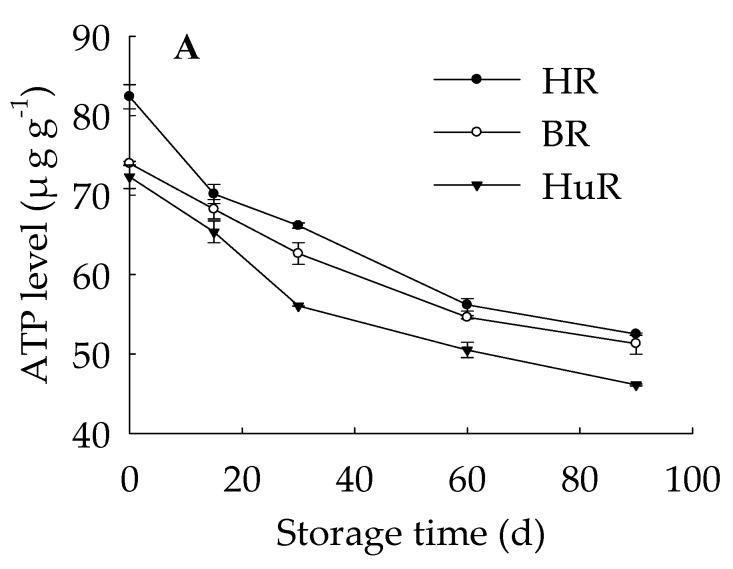
Changes in ATP (**A**), ADP (**B**), AMP (**C**) levels in juice sacs of three pumelo cultivars. Vertical bars represent SE of the means of three replicate assays.

**Figure 4 biomolecules-09-00701-f004:**
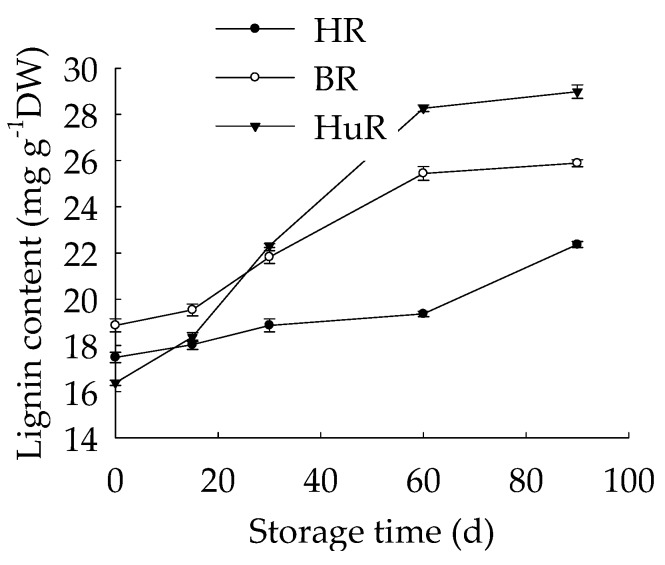
Changes in lignin content in juice sacs of three pumelo cultivars. Vertical bars represent SE of the means of three replicate assays. DW, dry weight.

**Figure 5 biomolecules-09-00701-f005:**
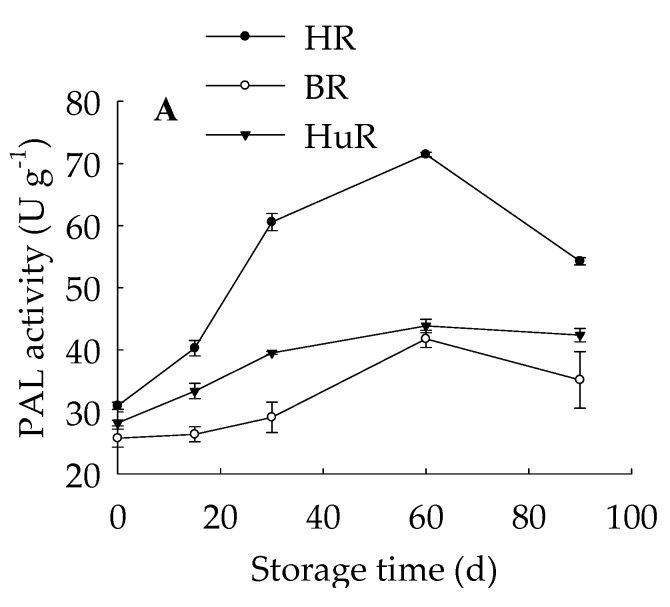
Changes in PAL (**A**), PPO (**B**), POD (**C**) activities in juice sacs of three pumelo cultivars. Vertical bars represent SE of the means of three replicate assays.

**Figure 6 biomolecules-09-00701-f006:**
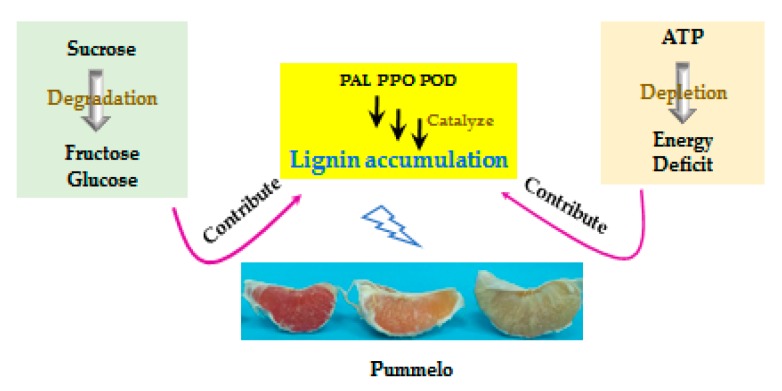
Probable mechanism of lignin accumulation in juice sacs of three pumelo cultivars by acting on the metabolism of sugar and energy.

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
