# Peer review of "Lignin Accumulation in Three Pumelo Cultivars in Association with Sucrose and Energy Depletion"

_biomolecules, 2019, doi:10.3390/biom9110701_

Round 1
Reviewer 1 Report
The manuscript describes the correlation of lignin accumulation to energy (ATP/ADP/AMP) and sucrose content. Overall the paper is well organized and makes reasonable correlations. I am skeptical of the use of acetyl bromide method of determining lignin here, as that is the only method used to determine lignin, and it is not clear that the authors are actually measuring lignin. An HSQC of the lignin structures and/or DFRC with corresponding mass spec would be preferable as this would prove the existence of lignin or at least lignin structures. However, the acetyl bromide method has been used in the literature and it is a reasonable assay to use. Overall, I suggest accepting the manuscript for publication once and HSQC is performed showing lignin structures (or some other secondary method for lignin verification such as DFRC or thioacidoloysis, not a Klason determination, is performed). Additionally, the authors should address the following with more detail:
It is weird that total soluble sugars for BR and HUR goes down from harvest to day 30 but then goes back up. Especially since this trend is not seen in glucose, sucrose, and fructose. What is causing this? Why? What sugars are getting depleted? Please comment/speculate.
On line 185, the authors describe the ATP/ADP/AMP changes as one of typical of physiological change. Please expand on this, how is this typical, where else has this been seen? This is particularly interesting given the fact that ADP sees a strange increase from day 60 to day 90, whereas ATP and AMP see a relatively steady decline (with the exception of AMP in BR).
Author Response
Reviewer #1:
The reviewers’ comments:
The manuscript describes the correlation of lignin accumulation to energy (ATP/ADP/AMP) and sucrose content. Overall the paper is well organized and makes reasonable correlations. I am skeptical of the use of acetyl bromide method of determining lignin here, as that is the only method used to determine lignin, and it is not clear that the authors are actually measuring lignin. An HSQC of the lignin structures and/or DFRC with corresponding mass spec would be preferable as this would prove the existence of lignin or at least lignin structures. However, the acetyl bromide method has been used in the literature and it is a reasonable assay to use. Overall, I suggest accepting the manuscript for publication once and HSQC is performed showing lignin structures (or some other secondary method for lignin verification such as DFRC or thioacidoloysis, not a Klason determination, is performed). Additionally, the authors should address the following with more detail:
The authors’ response:
Many thanks for the valuable suggestions or comments on our manuscript from the reviewer. As suggested, we will change a new method to analyze the lignin content such as DFRC or thioacidoloysis in the future.
The reviewers’ comments:
It is weird that total soluble sugars for BR and HUR goes down from harvest to day 30 but then goes back up. Especially since this trend is not seen in glucose, sucrose, and fructose. What is causing this? Why? What sugars are getting depleted? Please comment/speculate.
The authors’ response:
Thanks for the careful review, the soluble sugars are substrate of the respiration metbolism. The rapid respiration of farm fresh pumelo fruit consumes a lot of sugars, which may account for the decrease of the total soluble sugars. As the storag time extended, the respiration rate decreased so that sugar accumulated. As shown in Fig.1C, the sucrose content of BR and HuR decreased. There is a exception of HR in tendency of total soluble sugars and sucrose content, this is interesting and need further exploration.
The reviewers’ comments:
On line 185, the authors describe the ATP/ADP/AMP changes as one of typical of physiological change. Please expand on this, how is this typical, where else has this been seen? This is particularly interesting given the fact that ADP sees a strange increase from day 60 to day 90, whereas ATP and AMP see a relatively steady decline (with the exception of AMP in BR).
The authors’ response:
Thanks for the careful review, we considered the declined ATP level in postharvest pumlo storage was a typical of physiological change, because in the three pumelo cultivars we studies in our work all displayed the same characteristics. In addition, ATP declined not only in normal senescence but also in chilling injury process according to our previous studies (10.1016/j.scienta.2018.12.052). Energy dedicit was considered to one important factor that could result in fruit senescence and quality deterioration. ADP level went up from 60 d to 90 d, we speculated that it was a stress response to the energy depletion, since the engery level was low and various metabolism disorder occurred, the energy status could not be recovered in the late storage.

Reviewer 2 Report
The authors studied the physiological events and molecular actors associated with the post-harvest lignification process of juice sacs without segmental membranes (a sign of organoleptic deterioration), from three Chinese pomelos (Citrus maxima Merr) cultivars (HR, BR and HuR). Results on the change (0-90 days) in energy substrates (sugars, TSS) and related enzymes (invertases), energy balance (ATP / ADP / AMP), lignin and enzymes related to the production of phenolic compounds are reported. The authors argue that lignin production occurs through glucose channeling with a consequent negative energy balance in a cultivar-specific way (HuR> BR> HR).
Although there is indeed enough evidence to support the hypothesis, this reviewer considers that better statistical treatment of the experimental data could further unravel the causal relationship of the process. I suggest the following to improve the quality of research:
Major changes
Review the most correct common name for the studied fruit (is it pumelo, grapefruit or pomelo; https://www.gbif.org/species/3190160). Include a figure illustrating the biosynthetic pathway of lignin and related substrates in juice sacks during citrus granulation (doi: 10.1021 / jf5041349) and how this route could be connected with sugar and energy expenditure metabolism. Recent metabolomic data on this matter has been reported by other authors (doi: 1016/j.postharvbio.2016.03.010; 10.1016/j.postharvbio.2018.01.003; 10.1016/j.plantsci.2018.10.006; 10.3389/fpls.2019.00213). The authors could use simple and multiple correlation analysis, hierarchical analysis or principal component analysis (PCA) or PLS-DA to reveal the cultivar "individuality" of as well as the trends in metabolite production or consumption as related to enzyme fluctuations. For example, the analysis of changes in a single variety (e.g. jut HS) could reveal that lignin production is indeed related to the reduction of sugars (particularly glucose) or that ADP production between 60 or 90 days could be lately limiting glucose phosphorylation and POD activity in a dose-dependent manner. The simultaneous discussion of both, the differences for the same parameter between cultivars and the relationship of the substrate/metabolite/ enzyme mobilization, is quite confusing. I suggest discussing in a preliminary section all possible differences between cultivars (HS stands as the most differentiated sample). In many studies that unveil metabolic pathways, substrate or metabolite-based ratios are often used to discuss their "activation" or "inactivation" (for example, ATP / ADP + AMP or Glucose/sucrose). Re-construct the summary so that more quantitative information is included. All graphs should indicate statistical differences between treatments at a particular time-point.
Author Response
Review 2:
The reviewers’ comments:
The authors studied the physiological events and molecular actors associated with the post-harvest lignification process of juice sacs without segmental membranes (a sign of organoleptic deterioration), from three Chinese pomelos (Citrus maxima Merr) cultivars (HR, BR and HuR). Results on the change (0-90 days) in energy substrates (sugars, TSS) and related enzymes (invertases), energy balance (ATP / ADP / AMP), lignin and enzymes related to the production of phenolic compounds are reported. The authors argue that lignin production occurs through glucose channeling with a consequent negative energy balance in a cultivar-specific way (HuR> BR> HR).
Although there is indeed enough evidence to support the hypothesis, this reviewer considers that better statistical treatment of the experimental data could further unravel the causal relationship of the process. I suggest the following to improve the quality of research:
The authors’ response:
Thanks for the careful review and positive comment.
The reviewers’ comments:
Review the most correct common name for the studied fruit (is it pumelo, grapefruit or pomelo; https://www.gbif.org/species/3190160).
The authors’ response:
Thanks for the careful review, the studied fruit is pumelo (Citrus maxima (Burm.) Merr.).
The reviewers’ comments:
Include a figure illustrating the biosynthetic pathway of lignin and related substrates in juice sacks during citrus granulation (doi: 10.1021 / jf5041349) and how this route could be connected with sugar and energy expenditure metabolism.
The authors’ response:
Thanks for the careful review, the biosynthetic pathway of lignin is fully explored, and in this study we try to illustrate the relationship between lignin and sugar and energy metabolism. The possible mechanism was proposed as Figure 6.
The reviewers’ comments:
Recent metabolomic data on this matter has been reported by other authors (doi: 1016/j.postharvbio.2016.03.010; 10.1016/j.postharvbio.2018.01.003; 10.1016/j.plantsci.2018.10.006; 10.3389/fpls.2019.00213). The authors could use simple and multiple correlation analysis, hierarchical analysis or principal component analysis (PCA) or PLS-DA to reveal the cultivar "individuality" of as well as the trends in metabolite production or consumption as related to enzyme fluctuations. For example, the analysis of changes in a single variety (e.g. jut HS) could reveal that lignin production is indeed related to the reduction of sugars (particularly glucose) or that ADP production between 60 or 90 days could be lately limiting glucose phosphorylation and POD activity in a dose-dependent manner.
The authors’ response:
Thanks for the careful review and kind suggestions, we used correlation analysis to illustrate the relationship between lignin content and energy level and sugar content. The details are listed in the supplementary material.
The reviewers’ comments:
The simultaneous discussion of both, the differences for the same parameter between cultivars and the relationship of the substrate/metabolite/ enzyme mobilization, is quite confusing. I suggest discussing in a preliminary section all possible differences between cultivars (HS stands as the most differentiated sample). In many studies that unveil metabolic pathways, substrate or metabolite-based ratios are often used to discuss their "activation" or "inactivation" (for example, ATP / ADP + AMP or Glucose/sucrose).
The authors’ response:
Thanks for the careful review and kind suggestion, we discussed the possible differences between cultivars. In this work, we chose the results of BR to analyze the relationship of lignin accumulation and sucrose degradation and ATP depletion. Since the results of three pumelo cultivars were almost the same, we thought it would be reasonable to choose one cultivar to investigate the relationship of lignin and sucrose and ATP. The activities of enzymes in pumelo fruit during postharvest is changing but still active, "activation" or "inactivation" is not proper in our discussion.
The reviewers’ comments:
Re-construct the summary so that more quantitative information is included.
The authors’ response:
Thanks for the careful review, as suggested, we revised the text.
The reviewers’ comments:
All graphs should indicate statistical differences between treatments at a particular time-point.
The authors’ response:
Thanks for the careful review, all graphs indicated statistical differences between treatments and control, some statistical differences are too small to observe.

Reviewer 3 Report
The paper describes the lignin accumulation in juice sacs of three pumelo cultivars occurs during postharvest storage in regards to sucrose and energy depletion. Furthermore, it investigates the connection between the enhanced activities of three invertases with lignin accumulation during postharvestpumelo storage.
Abstract and Keywords: The abstract summarizes the essential information of the manuscript - how the research was carried out and major findings. However, a brief description of the methods used in this study is lacking altogether. I suggest the authors to include such brief description. Furthermore, it is not advised to use abbreviations as keywords, unless they are well established. In this case, NI, S-AI, B-AI and PAL may be mistaken for e.g. Phylogenetic Analysis Library (PAL) or Apolipoprotein (B/AI) or amylase inhibitor (AI). It would be better to use the full enzyme names as keywords.
The Introduction is well written. It gives the short literature review and describes the problems to be investigated. However, I suggest the authors to elaborate the aims of their study, i.e. to write the specific aims of the conducted study instead of the lines 59 -63. The sentence is too long and hard to follow.
The Materials and Methods section should be carefully read and rewritten where needed (see the line to line comments below). The work performed seems reasonable to address the topic, although a bit more information needs to be given in parts of the methods.
Although the units denoted as “mL, L” for volume are commonly used, the authors should change it to SI unit, as requested in the Instructions to Authors. The same should be applied to all units throughout the manuscript.
The manuscript as a whole is written in fair English. However, the Materials and Methods section would benefit greatly by review from a native English speaker.
Line 70 : Where they were stored and how did you keep the 85% relative humidity?
Line 75: Describe the grinding procedure – what mill was used, what was the particle size of sample after grinding?
Line 79: ALTUS A-10 (Perkin Elmer)
Line 82: “The sugar content….from the sugar standard curve” (Only one or standard curve for each sugar determined)
Lines 82-83: sugar contents were expressed as mg g-1
Line 83:”identification of sugars were calculated” What is the meaning of this?
Line 84: Please, edit the equation (font size, subscripts)
Line 89: “Five grams of juice sacs” – prepared as described in section 2.2 (powder) or juice sacs in their native form ? Please, specify.
Lines 97-102: The assays are not clearly described. The reagents are listed, but volumes of sample and reagents are not given. How the assays were performed – well-plates or else? How the enzyme activities were calculated and expressed? Please, rewrite this section.
Line 117: “A total of 1 g of juice sacs…” prepared as described in section 2.2? Please, specify.
Line 127: Please, specify the concentration of acetic acid.
Lines 132-133: “Significant difference…” – the sentence is completely incomprehensible. Please, rewrite or paraphrase it.
The results are presented clearly and in an organized form (figures) and adequately explained.
The titles of the Results subsections 3.1. to 3.5. are identical (Changes in antioxidant ability in juice sacs of three pummelo cultivars). Please, correct it and write the appropriate titles.
The figures are adequately numbered, sufficiently self-explanatory and accompanied by captions. However, the explanation of what is represented by figure A, B, C and D should be included in the caption of Figure 1.
The results are well presented and elaborated in the Discussion section and the previous work (of other authors) in the field appropriately presented and compared to the results obtained with this study.
Line 295: “Since the differences of the above parameters among…were not very large…” It would be good to state whether the differences were statistically significant or not.
The conclusion is adequately derived from the obtained results.
Line 348-349: The last sentence in the manuscript, along with the Figure 6, would be more appropriate for the last subsection of the Discussion. It would be very good if the authors describe the mechanism given in Figure 6 in more details.
The references are given in the text according to the Instructions to Authors, they are listed in the References section in the order in which they occur in the text and they comply with the citation style given in the Instructions.
Author Response
Review 3:
The reviewers’ comments:
The paper describes the lignin accumulation in juice sacs of three pumelo cultivars occurs during postharvest storage in regards to sucrose and energy depletion. Furthermore, it investigates the connection between the enhanced activities of three invertases with lignin accumulation during postharvest pumelo storage.
The authors’ response:
Many thanks for the valuable suggestions or comments on our manuscript from the reviewer.
The reviewers’ comments:
Abstract and Keywords: The abstract summarizes the essential information of the manuscript - how the research was carried out and major findings. However, a brief description of the methods used in this study is lacking altogether. I suggest the authors to include such brief description. Furthermore, it is not advised to use abbreviations as keywords, unless they are well established. In this case, NI, S-AI, B-AI and PAL may be mistaken for e.g. Phylogenetic Analysis Library (PAL) or Apolipoprotein (B/AI) or amylase inhibitor (AI). It would be better to use the full enzyme names as keywords.
The authors’ response:
Thanks for the careful review, as suggested, we have added a brief description of methods, and revised the keywords.
The reviewers’ comments:
The Introduction is well written. It gives the short literature review and describes the problems to be investigated. However, I suggest the authors to elaborate the aims of their study, i.e. to write the specific aims of the conducted study instead of the lines 59 -63. The sentence is too long and hard to follow.
The authors’ response:
Thanks for the careful review, as suggested, we have revised the lines 59-63.
The reviewers’ comments:
The Materials and Methods section should be carefully read and rewritten where needed (see the line to line comments below). The work performed seems reasonable to address the topic, although a bit more information needs to be given in parts of the methods.
The authors’ response:
Thanks for the careful review, as suggested, we have revised the Materials and Methods section.
The reviewers’ comments:
Although the units denoted as “mL, L” for volume are commonly used, the authors should change it to SI unit, as requested in the Instructions to Authors. The same should be applied to all units throughout the manuscript.
The authors’ response:
Thanks for the careful review, we have revised it.
The reviewers’ comments:
The manuscript as a whole is written in fair English. However, the Materials and Methods section would benefit greatly by review from a native English speaker.
The authors’ response:
Thanks for the careful review, we have revised it.
The reviewers’ comments:
Line 70 : Where they were stored and how did you keep the 85% relative humidity?
The authors’ response:
The fruit were stored at a storehouse and the 70%-85% relative humidity was controlled by a humidifier.
The reviewers’ comments:
Line 75: Describe the grinding procedure – what mill was used, what was the particle size of sample after grinding?
The authors’ response:
The sample frozen in liquid nitrogen was weighed and ground with a ginder, and we consider different samples were under the same treatment because all samples were kept the same weight and the same grinding time. The particle size of sample after grinding was not measured.
The reviewers’ comments:
Line 79: ALTUS A-10 (Perkin Elmer)
The authors’ response:
Thanks for the careful review, we have revised it.
The reviewers’ comments:
Line 82: “The sugar content….from the sugar standard curve” (Only one or standard curve for each sugar determined)
The authors’ response:
Thanks for the careful review, each sugar has one standard curve.
The reviewers’ comments:
Lines 82-83: sugar contents were expressed as mg g-1
The authors’ response:
Thanks for the careful review, we have revised it.
The reviewers’ comments:
Line 83:”identification of sugars were calculated” What is the meaning of this?
The authors’ response:
Thanks for the careful review, we have revised the sentence as “The contents of sugars were calculated using the following equation”
The reviewers’ comments:
Line 84: Please, edit the equation (font size, subscripts)
The authors’ response:
Thanks for the careful review, we have revised the equation.
The reviewers’ comments:
Line 89: “Five grams of juice sacs” – prepared as described in section 2.2 (powder) or juice sacs in their native form ? Please, specify.
The authors’ response:
Thanks for the careful review, we have revised it.
The reviewers’ comments:
Lines 97-102: The assays are not clearly described. The reagents are listed, but volumes of sample and reagents are not given. How the assays were performed – well-plates or else? How the enzyme activities were calculated and expressed? Please, rewrite this section.
The authors’ response:
Thanks for the careful review, the assays were performed in well-plates. As suggested, we have revised the text.
The reviewers’ comments:
Line 117: “A total of 1 g of juice sacs…” prepared as described in section 2.2? Please, specify.
The authors’ response:
Thanks for the careful review, we have revised the text.
The reviewers’ comments:
Line 127: Please, specify the concentration of acetic acid.
The authors’ response:
Thanks for the careful review, we have added the concentration of acetic acid.
The reviewers’ comments:
Lines 132-133: “Significant difference…” – the sentence is completely incomprehensible. Please, rewrite or paraphrase it.
The authors’ response:
Thanks for the careful review, we have revised the text.
The reviewers’ comments:
The results are presented clearly and in an organized form (figures) and adequately explained.
The authors’ response:
Thanks for your careful review and the positive comment.
The reviewers’ comments:
The titles of the Results subsections 3.1. to 3.5. are identical (Changes in antioxidant ability in juice sacs of three pummelo cultivars). Please, correct it and write the appropriate titles.
The authors’ response:
Thanks for your careful review, we have revised the text.
The reviewers’ comments:
The figures are adequately numbered, sufficiently self-explanatory and accompanied by captions. However, the explanation of what is represented by figure A, B, C and D should be included in the caption of Figure 1.
The authors’ response:
Thanks for your careful review, we have revised the text.
The reviewers’ comments:
The results are well presented and elaborated in the Discussion section and the previous work (of other authors) in the field appropriately presented and compared to the results obtained with this study.
The authors’ response:
Thanks for your careful review and the positive comment.
The reviewers’ comments:
Line 295: “Since the differences of the above parameters among…were not very large…” It would be good to state whether the differences were statistically significant or not.
The authors’ response:
Thanks for your careful review, we have revised the text.
The reviewers’ comments:
The conclusion is adequately derived from the obtained results.
The authors’ response:
Thanks for your careful review and the positive comment.
The reviewers’ comments:
Line 348-349: The last sentence in the manuscript, along with the Figure 6, would be more appropriate for the last subsection of the Discussion. It would be very good if the authors describe the mechanism given in Figure 6 in more details.
The authors’ response:
Thanks for your careful review, as suggested, we have revised the text.
The reviewers’ comments:
The references are given in the text according to the Instructions to Authors, they are listed in the References section in the order in which they occur in the text and they comply with the citation style given in the Instructions.
The authors’ response:
Thanks for your careful review and the positive comment.
